# Application of Miniature FBG-MEMS Pressure Sensor in Penetration Process of Jacked Pile

**DOI:** 10.3390/mi11090876

**Published:** 2020-09-21

**Authors:** Xueying Liu, Yonghong Wang, Mingyi Zhang

**Affiliations:** 1School of Civil Engineering, Qingdao University of Technology, Qingdao 266033, China; liuxueying@hnu.edu.cn (X.L.); zhangmingyi@qut.edu.cn (M.Z.); 2Collaborative Innovation Center of Engineering Construction and Safety in Shandong Blue Economic Zone, Qingdao University of Technology, Qingdao 266033, China

**Keywords:** jacked pile, FBG test technology, MEMS micro-sensing technology, penetration mechanism, pile stress, pile–soil interface

## Abstract

In order to study the penetration mechanism of jacked piles in viscous soil foundation, the stress variation law of the pile–soil interface was obtained by installing silicon piezoresistive earth pressure and pore water pressure sensors, and fiber Bragg grating (FBG) sensors in a model pile body, and the penetration characteristics of jacked piles in homogeneous viscous soil were defined. The test results show that: Fiber Bragg grating and silicon piezoresistive sensing technology can better meet the requirements of testing the characteristics of jacked pile in viscous soil. The ratio of pile lateral resistance to pile end resistance varies when pile is jacked in homogeneous viscous soil. In the early stage of pile jacking, the ratio of pile lateral resistance is small, and in the later stage of pile jacking, the ratio of pile lateral resistance increases, but the ratio of pile end resistance is still higher than that of pile lateral resistance. The ratio of the effective stress to the total radial stress is high, and the variation law of the two is consistent with the depth. The total radial stress, pore water pressure, and effective radial stress all exhibit the degradation phenomenon, and the degradation degree decreases gradually with the increase in penetration depth at the same depth. The ratio of excess pore water pressure to overburden weight decreases with the increase in depth, and the maximum value is 87%. The research results can provide a reference for the engineering practice of jacked pile in viscous soil foundation.

## 1. Introduction

With the continuous development of urban construction and the gradual improvement in jacked pile technology, jacked pile is more widely used in soft soil areas and densely populated cities [1,2,3,4]. In recent years, fiber Bragg grating (FBG) technology has been continuously developed and has gradually become an effective test method for civil engineering health monitoring, and many beneficial results have been achieved [5,6,7,8]. Therefore, how to obtain the mechanism of jacked pile penetration in viscous soil foundation has become a hot topic in the research of jacked pile [9,10].

Scholars at home and abroad have conducted a series of beneficial studies on the mechanism of jacked pile penetration: Yang et al. [11] studied the influence of different pile jacking methods on the performance of H pile. The penetration mechanism of jacked pile is obtained by installing the instrument pile in residual soil, which is similar to silt. Togliani [12] and Igoe et al. [13] prestressed steel bars embedded in prefabricated piles and discussed the associated testing problems. Oh et al. [14] implanted a quasi-distributed optical fiber sensor in the pipe pile grooves to separate the side friction resistance and pile end resistance of the jacked pile penetration process to study the timeliness of the bearing capacity. Lee et al. [15] used the Brillouin method of optical fiber sensing to realize a distributed fiber strain test for prefabricated piles. Klar et al. [16] compared the results of the single pile static load test using either Brillouin distributed fiber technology or ordinary sensors and expounded the economics of the two approaches. Pando et al. [17] performed experimental research on the process to embed a fiber grating sensing system in a jacking pile, which solved the problem of implanting a fiber FBG sensor in the pipe pile. Weng et al. [18] established a sensor network based on a fiber Bragg grating to monitor the strain distribution in a pavement structure under different settlement conditions. The improved packaging and installation method of the quasi-distributed sensor system not only guarantees the high survival rate of the sensor, but also realizes the accurate measurement of longitudinal and transverse axial strain.

The traditional method to measure the earth and pore water pressures during the jacking process and the rest period after jacking is to embed earth and pore water pressure gauges in the affected soil layer at certain distances around the pile. Pestana et al. [19] embedded earth and pore water pressure gauges in the soil around the pile to explore pressure changes in the soil at the pile end and certain distances around the pile during pile jacking. McCabe et al. [20] buried earth and pore water pressure sensors in the soil around the pile and presented a case history describing the measurements collected during the installation and load testing of groups of five, closely spaced, precast, concrete piles in a soft clay-silt. The earth and pore water pressures at the pile–soil interface act directly on the pile body, which is more influential than when further from the pile. However, this kind of test has been reported in the literature few times [21], which limits the in-depth study of the pile–soil interface characteristics and bearing capacity mechanism.

At present, comprehensive research on the mechanism of jacked pile penetration is not deep enough, and the relevant characteristics obtained are not accurate enough. The applicability of FBG sensing technology and miniature silicon piezoresistive sensing technology in obtaining the mechanism of jacked pile penetration has not been accurately verified. Based on this, the penetration mechanism of jacked pile in viscous soil foundation was studied by using silicon piezoresistive pressure sensors and FBG fiber Bragg grating sensors, and a special nylon rod model pile. The axial force of pile and the resistance of pile jacking are analyzed, and the proportional relation between pile end resistance and pile lateral resistance is further studied. The change in the total radial stress and pore water pressure along the depth direction in the process of pile jacking is deeply analyzed. It is of practical significance to guide the design of jacked pile engineering.

## 2. Development of Miniature Silicon Piezoresistive Sensor

### 2.1. Working Principle of Silicon Piezoresistive Pressure Sensor

The silicon piezoresistive pressure sensor adopts the advanced miniaturization manufacturing process to integrate the silicon pressure diaphragm as the sensing element, and takes advantage of the piezoresistive effect of polysilicon to prepare four polysilicon varistors by deposition on the insulating layer of silicon dioxide deposited on the pressure diaphragm to form the Wheatstone bridge [22,23]. A schematic diagram of the silicon piezoresistive pressure sensor is shown in Figure 1, and the circuit of the Wheatstone bridge is shown in Figure 2.

In this paper, the Wheatstone bridge adopts a constant voltage source power supply. The pressure difference between the two sides of the silicon diaphragm makes the resistance value of the four resistors above also change. The Wheatstone bridge is out of balance, and the output voltage [24] is:(1)$$V_{0}=\frac{\left[(R_{1}+\Delta R_{1})(R_{3}+\Delta R_{3})-(R_{2}-\Delta R_{2})(R_{4}-\Delta R_{4})\right]}{\left(R_{1}+R_{2}+\Delta R_{1}-\Delta R_{2})(R_{3}+R_{4}+\Delta R_{3}-\Delta R_{4}\right)}\times V_{B}$$
where $V_{B}$ is the supply voltage and $V_{0}$ is the output voltage. If $R_{1}=R_{3}=R_{2}=R_{4}=R$, then $\Delta R_{i}=R\cdot GF\cdot \varepsilon_{i}$ (i=1,2,3,4), and $\varepsilon_{i}$ is the strain value of the *i*th resistance as follows [25]: (2)$$V_{0}=\frac{1}{4}GF\cdot \frac{\varepsilon_{1}+\varepsilon_{3}-\varepsilon_{2}-\varepsilon_{4}}{\left[1+\frac{1}{2}(\varepsilon_{1}+\varepsilon_{2}+\varepsilon_{3}+\varepsilon_{4})\right]}V_{B}$$

When the sensor is designed, the four resistance strain values satisfy $\varepsilon_{1}=\varepsilon_{3}=-\varepsilon_{2}=-\varepsilon_{4}=\varepsilon$, and Equation (2) becomes: (3)$$V_{0}=GF\cdot \varepsilon \cdot V_{B}$$

By Equation (3), the voltage generated by the Wheatstone bridge has a linear relationship with the strain value on the varistor, and the output voltage can be measured under external pressure. The voltage value is related to the gauge factor of the diaphragm. The greater the gauge factor and strain value of the diaphragm, the greater the pressure value output by the bridge, and the higher the sensitivity of the sensor.

### 2.2. Design of Silicon Piezoresistive Pressure Sensor

The miniature silicon piezoresistive pressure sensor adopts the advanced miniaturization manufacturing process to integrate the silicon diaphragm as the sensitive element. The four resistors on the diaphragm are connected through the screen-printing circuit, and the delicate micro-packaging is carried out with the metal shell [26,27]. A schematic diagram of the sensor packaging structure is shown in Figure 3, and the picture of the sensor is shown in Figure 4. According to the test requirements of the jacked pile penetration model test, the silicon piezoresistive earth pressure sensor and the silicon piezoresistive pore water pressure sensor were used. The surface of the pore water pressure sensor is sealed with permeable stone. Due to the size limitation of the model pile, different outgoing methods are adopted according to the installation process of the sensor on the model pile.

The calibration test results show that all the silicon piezoresistive pressure sensors have a good linear fitting degree and the linear correlation coefficient is above 0.99. The specific performance indexes of the miniature silicon piezoresistive pressure sensors are shown in Table 1.

## 3. Development of Sensitized FBG Strain Sensor

### 3.1. Working Principle of Fiber Bragg Grating

The fiber Bragg grating makes use of the photosensitivity of germanium, phosphorus, and other optical fiber materials to write incident light to the fiber core and interact with its doped particles, resulting in periodic changes in the refractive index of the fiber core. A grating with spatial phase is formed in the fiber core, and multiple gratings can be written simultaneously to achieve distributed sensing. As shown in Figure 5, the light and dark of the core in the figure alternately indicate the periodic change in the refractive index [28], Λ is the grating period, and the wavelength shift of the FBG has the following relationship with the FBG axial strain and temperature change: (4)$$\frac{\Delta \lambda_{b}}{\lambda_{b}}=\eta \varepsilon +\gamma \left(T-T_{0}\right)$$
where $\frac{\Delta \lambda_{b}}{\lambda_{b}}$ is the wavelength change rate of the fiber grating; η is the strain coefficient; γ is the temperature coefficient; ε is the axial strain of the fiber; and $\left(T-T_{0}\right)$ is the change in temperature.

The temperature change in the laboratory test is not big and can be ignored. Therefore, when the external temperature change is not taken into account, the sensitivity coefficient $K_{\varepsilon }=\eta \lambda_{b}$ of the strain sensing of the fiber Bragg grating can be obtained as follows: (5)$$\Delta \lambda =K_{\varepsilon }\cdot \varepsilon$$

### 3.2. Design and Parameters of Sensitized FBG Sensor

The working principle structure of the sensitized FBG strain sensor is shown in Figure 6. The sensor is composed of a fiber Bragg grating, two clamping bushings, and a tail fiber. The fiber at both ends of the fiber grating is fixed in the clamping tube with a binder to avoid the influence of the binder on the strain transfer of the fiber grating [29]. The sensor can be either fixed to the test member by clamping support or embedded in the structure. With the hypothesis test component between the two clampings bearing axial deformation ΔL, and the corresponding clamping deformation of the casing and fiber grating $\Delta L_{s}$ and $\Delta L_{f}$, respectively, optical fiber and clamping deformation of the casing binder does not occur, the sensor in the clamping casing deformation can be ignored, clamping between the bearing deformation is produced by the fiber Bragg grating, and the relationship between deformation and testing components for the sensor is [30]: (6)$$L\varepsilon =L_{f}\varepsilon_{f}$$

When the center wavelength of the optical fiber core of the sensor is pure quartz at the band of 1550 nm, $K_{\varepsilon }\approx 1.2\;\mathrm{pm}/\mu \varepsilon$, according to Equations (5) and (6),
(7)$$\varepsilon =\frac{L_{f}}{L}\varepsilon_{f}=\frac{L_{f}\Delta \lambda_{FBG}}{1.2L}$$
where *L_f_* is the distance of the fiber Bragg grating (mm) between the clamping sleeve; *L* is the distance between two clamping supports (mm); *ε_f_* is the strain of the fiber Bragg grating between clamping casings; and Δ*λ_FBG_* refers to the wavelength change of the fiber grating (nm).

According to Equation (7), the sensitivity coefficient of the sensor can be adjusted by changing the ratio of $L_{f}$ to L.

The picture of the sensitized FBG strain sensor is shown in Figure 7. It can be seen from the figure that the clamping sleeve on both sides adopts a 1.7 mm stainless steel tube, and the black interior of the sensor is the position of the fiber grating. With the fiber grating standard distance of 23 mm, the fiber core adopts a naked fiber grating, and the tail fiber adopts a 0.9 mm loose sleeve protection. The relevant parameters of the FBG strain sensor are shown in Table 2.

## 4. Overview of Model Test

### 4.1. Model Test Barrel

The model barrel used in the test was welded from a 1.5 mm-thick iron plate, with a diameter of 800 mm and a height of 1200 mm. The stirrup was welded in the middle of the model bucket to limit the deformation of the bucket body. The guide device was welded in the middle of the section to ensure the pile body was vertical during the process of model pile penetration. Three scales were pasted into the bucket to facilitate the making of the model foundation in the later period.

### 4.2. Soil Sample Preparation

According to the test requirements, the silty clay foundation of a construction site in Qingdao was selected to make the model foundation. The soil samples retrieved from the site were taken back to the laboratory, and the moisture content of the model foundation was set to 28%. The model foundation was made in layers in the model bucket with a thickness of 10 cm per layer. The quality of dry soil and water required for each layer of foundation was calculated, and the dry soil was evenly put into the model bucket. With the help of a scale in the bucket, the hammer was rammed to the design height and smoothed, and the wetting was uniform. After completion, the opening of the barrel was sealed with a plastic sheet and left to sit for 24 h to allow the water to fully soak. After the foundation was finished, the mouth of the model bucket was sealed with plastic cloth for 7 days to make the water in the foundation soil sample even. The soil sample preparation process is shown in Figure 8. Before the laboratory test, the prepared foundation soil was taken for geotechnical test, and its physical and mechanical indexes were measured, as shown in Table 3.

### 4.3. Model Pile

In this test, nylon rod was used as the model pile. Considering the boundary effect [31] and the size of the model barrel, the size of the designed model pile was: 60 mm in diameter and 750 mm in length. Effective pile length was 700 mm. According to the similarity principle, the geometric similarity ratio was about 1:10. The elastic modulus of nylon rod is about 2.8 GPa, the elastic modulus of C60 prestressed pipe pile commonly used in engineering is about 36 GPa, and the model ratio of elastic modulus is about 1:12. The test model pile adopted a solid nylon rod, without considering the plugging effect.

### 4.4. Test Elements

In order to obtain the stress at the pile–soil interface in the process of pile jacking, the silicon piezoresistive earth pressure sensor, pore water pressure sensor, and FBG sensor were installed at different sections of the model pile. Each group was installed at the same cross-section, and 5 silicon piezoresistive earth pressure sensors, pore water pressure sensors, and FBG sensors were used. The FBG sensor was 12 mm in height and 20 mm in diameter. The relevant parameters of the earth pressure sensor and the water pressure sensor are shown in Table 4. The distribution diagram and installation process of the sensors are shown in Figure 9 and Figure 10.

### 4.5. Loading System

The reaction frame and hydraulic jack were used as the loading device. The jack was fixed on the beam of the reaction frame by a steel strand, and the model barrel was placed under the reaction frame. The model pile was continuously penetrated into the soil at a speed of 300 mm/min. The test device is shown in Figure 11.

## 5. Test Results and Analysis

### 5.1. Penetration Resistance

The pile lateral resistance was calculated by measuring the pile force and pile end resistance respectively by the high-frequency pressure sensor on the top surface of the loading device and the miniature silicon piezoresistive-type earth pressure sensor mounted on the pile end. The curve of the three changes with the penetration depth can be drawn, as shown in Figure 12.

It can be seen from Figure 13 that in homogeneous clay, the pile pressure increases with the penetration depth, and instead of a simple linear increase, it is a volatile increase. When the penetration depth of the model pile is within the range of 0–6*D* (*D* is the diameter of the model pile), the pile pressure increases greatly at the initial breaking of soil, and then the pile pressure is relatively stable with little change. This is mainly because when the soil layer is made, there is a hard shell on its surface. When the soil is broken into, the pile end resistance increases greatly, which leads to the increase in pile pressure. As the penetration depth continues to increase, the shear strength of the soil significantly decreases, and the pile lateral resistance is small, resulting in a small increase in pile pressure. The maximum pile pressure within this range is 0.845 kN. When the penetration depth reaches 6*D*, the pile pressure continues to increase after a small reduction, with a maximum increase of 22.4%. This is mainly because with the increase in depth, the soil strength around the pile gradually recovers, and the pile lateral resistance begins to increase, so the pile pressure increases. As the depth continues to increase, the pile pressure gradually tends to be constant. Within this depth range, the piling force is 0.711–1.304 kN.

When the penetration depth is less than 2*D*, the pile end resistance increases sharply, increasing by 29.2%, and the growth rate along the penetration depth is large. This is mainly because in the early stage of pile jacking, the penetration of the pile end will produce greater resistance. When the penetration depth is 2*D*–8*D*, the pile end resistance increases gently with penetration depth, with an increase range of 7.4–10.5% and an increase rate of 1.63–3.26% along the depth, indicating that the pile lateral resistance gradually increases, but the pile end resistance still occupies a large proportion of pile pressure. When the penetration depth exceeds 8*D*, the pile end resistance increases with the penetration depth, with a growth range of 36.5–97.8% and a growth rate of 1.23–3.3%. It indicates that the pile lateral resistance increases rapidly in the deeper soil layer. It can be seen that even in homogeneous soil, pile end resistance is not linearly increased, but shows an increase in volatility. The ratio of pile end resistance is not a fixed value. At the initial penetration, the penetration depth of the pile is small, the contact area between the pile side and the soil is small, and the surface soil is subject to radial disturbance. At this time, most of the pile pressure is borne by the pile end resistance. With the continuous penetration of the pile body, the ratio of pile lateral resistance to pile pressure is increased in the middle and later period.

When the penetration depth is less than 6*D*, the pile lateral resistance is rapid, but the value is small. This is mainly because the surface area of the pile side contacting with the soil is small, and the value is small at the initial penetration stage because the pile body is shaking and the contact between the pile and the soil is not close. When the penetration depth is 6*D*–8*D*, the pile lateral resistance continues to increase, with an increase of 21.6–29.6%, and the growth rate continues to increase. When the penetration depth is more than 8*D*, the pile lateral resistance continues to increase with the penetration depth, with a larger growth range and a larger growth rate, and the ratio of pile lateral resistance to pile pressure increases. In general, there is a weak degradation of pile lateral resistance at the same depth, and the degradation rate decreases gradually along the direction of depth. This is mainly due to the shaking of the pile, and the stress release of the shallow soil is greater in the process of penetration, so the degradation degree of the upper soil is greater than that of the lower soil, but the velocity slows down along the direction of depth.

### 5.2. Pile Axial Force

Pile strain can be obtained according to the measured data of FBG sensors. Pile stress and pile axial force during pile jacking can be obtained from pile strain. The calculation equations are as follows: (8)σ=E⋅Δε
(9)N=σ⋅A
where *σ* is the pile stress (kPa); *E* is the elastic modulus of pile (kPa); Δ*ε* is the strain variation of pile; *N* is the pile axial force (kN); *A* is the cross-sectional area of the pile (m^2^).

According to the data measured by the installed miniature FBG sensor, the pile axial force is calculated and the curve of the axial force changing with the depth is drawn, as shown in Figure 13.

It can be seen from Figure 13 that the pile axial force decreased with the increase in the penetration depth of pile. At the same penetration depth, the pile axial force decreased with the increase in the penetration depth, indicating the degradation of pile lateral resistance. In the process of penetration of 20 cm, the pile axial force decreased rapidly, with a decrease range of 70.3%, indicating that the pile axial force fluctuated greatly in penetration depth, so the pile lateral resistance increased. In the process of penetration of 35 cm, with the penetration depth increasing, the pile axial force decreased by 21.7–46.8%, the reduction rate decreased, and the pile lateral resistance at the same depth deteriorated. In the process of penetration of 50 cm, the decrease in the pile axial force was 10.2% to 32.9%. The decrease in the axial force of the pile body continued to decrease, the degree of degradation of the pile lateral resistance increased, and the lateral resistance continued to decrease. In the process of penetration depth of 70 cm, when the penetration depth was less than 7*D*, the pile axial force decreased slowly, only 4.8–7.2%. Within this range, the degradation in pile lateral resistance was more significant, so the reduction in the pile axial force is smaller. With the increase in depth, in the range of more than 7*D*, the decrease in the pile axial force increased from 10.5% to 14.2%. At this depth, the degradation in the pile lateral resistance was not as significant as that of the upper part. Generally speaking, the pile axial force gradually decreased with the penetration depth, the pile axial force at the same penetration depth gradually decreased with the increase in the depth, and the decrease gradually decreased. It was shown that the soil around the pile that had penetrated the interface gradually strengthened the radial restraint during the penetration of the pile jacking.

### 5.3. Total Radial Stress

Through the data measured by the installed silicon-piezoresistive earth pressure sensor, the curve of the total radial stress at the pile–soil interface with the penetration depth can be obtained, as shown in Figure 14.

As can be seen from Figure 14, with the penetration depth of the sensor increasing gradually, the total radial stress increased gradually. The reason is that the foundation soil of this test was homogeneous soil, and the increase range of radial stress was relatively stable. The curve *H*/*L* = 1/14 was the radial stress measured by the sensor at the bottom, which can be considered as the radial stress without degradation. It can be seen that the radial stress increased significantly at the initial penetration, mainly because the soil layer was a saturated clay with high cohesion. With the increase in penetration depth, the radial stress increased steadily, reaching 14.9 kPa at 45 cm. As the depth continued to increase, the increase amplitude of the total radial stress decreased. This is mainly due to the film coating effect of the viscous soil on the sensor surface in the test. As the penetration depth increased, the film coating effect became more significant, so the total radial stress decreased. According to the variation law of the total radial stress measured by other sensors, it can be found that the total radial stress had the degradation phenomenon, and the degradation degree decreased gradually with the increase in the sensor height. At 10 cm, the degradation degree decreased from 5.86 to 3.68 kPa and then to 2.93 kPa with the increase in the sensor height. The reduction ranges were 37.2% and 25.3%, respectively, and the degradation degree gradually decreased. This is mainly because when the pile end is cut into, the clay layer will have a “well wall” effect, which reduces the degradation degree of the total radial stress.

### 5.4. Analysis of Pore Water Pressure at Pile–Soil Interface

#### 5.4.1. Pore Water Pressure

According to the data measured by the pore water pressure sensor installed in the pile, the change curve of pore water pressure at the pile–soil interface with the penetration depth can be obtained, as shown in Figure 15.

As can be seen from Figure 15, in the process of pile jacking, the pore water pressure measured by each silicon piezoresistive pore water pressure sensor gradually increased with the increase in the depth of the sensor, and its trend was approximately linear. This is mainly because the foundation soil of this laboratory test was homogeneous soil, and the increase range of pore water pressure was basically the same with the increase in depth, so the distribution law of pore water pressure was linear. At the same time, due to the shear disturbance of the pile to the soil, the deep soil was more closely contacted with the surface of the pile, which led to the continuous increase in pore water pressure in the deeper part. When pile jacking depth was small, the overburden of soil around the pile was small, the horizontal lateral pressure was small, and the pore water pressure was small. With the increase in pile jacking depth, the horizontal pressure increased, and the pore water pressure generated by pile–soil shear was larger. As the position of the sensor moved up, the measured pore water pressure decreased. This is because the deeper the pile sinks, the longer the shear action of the pile lasts, making the pore water rise. However, the continuous disturbance of the soil around the pile also provided a channel for the dissipation of the pore water, resulting in the reduction in the pore water pressure to some extent, but very weak. This is because the deeper the pile jacks, the longer the shear action of the pile lasts, making the pore water rise. However, the continuous disturbance of the soil around the pile also provided a channel for the dissipation of the pore water, resulting in the reduction in the pore water pressure to some extent, but very weak.

#### 5.4.2. Excess Pore Water Pressure

The excess pore water pressure is the difference between the measured pore water pressure and the static water pressure, and its curve changes with penetration depth, as shown in Figure 16.

As can be seen from Figure 16, the excess pore water pressure fluctuated with the increase in the pile depth, which is similar to the distribution law of the excess pore water pressure in the study of Juan et al. [19]. According to the curve obtained by the sensor at *H*/*L* = 1/14, the excess pore water pressure was larger at the initial penetration, and increased with the penetration depth. This is mainly because with the increase in penetration depth, the pile–soil interface is gradually compact, and the pore water pressure generated in the process of pile jacking dissipates slowly, leading to the gradual rise in the excess pore water pressure. In addition, according to the comparison of data at the same depth, it can be found that the excess pore water pressure dissipated, and the dissipation range decreased with the increase in the depth. At 15 cm, the excess pore water pressure dissipated by 15.7% and at 40 cm by only 3.9%. This is mainly because the shallow soil had undergone the maximum degree of shear action in the process of pile jacking, the structure damage degree was large, and the pore water dissipated quickly, so the excess pore water pressure dissipated quickly in the upper part.

As the pore water pressure sensor *H*/*L* = 1/14 is located at the end of the pile, it can approximately represent the distribution of the excess pore water pressure along the pile body during the whole pile jacking process. Thus, the ratio of the excess pore water pressure of the model pile at different depths to the effective overlying soil weight can be obtained as shown in Table 5.

It can be seen from Table 5 that the excess pore water pressure of model pile is always large in the process of pile jacking, and the ratio reaches 84.7% at 30 cm. However, with the increase in depth, the proportion of excess pore water pressure decreases gradually, and the ratio has decreased to 38.7% at 70 cm. The reason is that with the increase in penetration depth, the growth range of excess pore water pressure decreases, resulting in the ratio of excess pore water pressure to overburden weight decreasing gradually. The excess pore water pressure has a great influence on the bearing capacity of pile foundation. Therefore, in practical engineering, some measures should be taken to prevent the excess pore water pressure from affecting the bearing capacity of pile foundation during the pile jacking process of saturated clay foundation soil.

### 5.5. Effective Radial Stress

According to the total radial stress and the excess pore water pressure, the curve of the radial effective stress at the pile–soil interface with the penetration depth can be obtained, as shown in Figure 17. According to the change curves of total radial pressure and effective radial pressure with penetration depth, the ratio of effective radial stress at different depths to total radial stress can be obtained, as shown in Table 6.

As can be seen from Figure 17, the radial effective stress in the process of pile jacking increased with the penetration depth, and increased linearly. It can be seen from Table 6 that the radial effective stress is the main component of the total radial stress, accounting for 78.9–89.4%, which is similar to the distribution law of the total radial stress. By comparing the curves obtained by sensors at different positions, it can be found that the radial effective stress degraded with the increase in penetration depth at the same depth. This is because the repeated shear action of the same depth gradually intensified in the process of pile jacking, the pile–soil fitting degree gradually decreased, and the radial effective stress degraded. The degradation law of radial effective stress is similar to that of total radial stress. As the height of the sensor increased, the degree of degradation gradually decreased. This is mainly due to the decrease in the effective radial stress after the pile tip shear penetration, but with the increase in the penetration depth, the decrease in the effective radial stress decreased.

## 6. Conclusions

(1)Fiber grating sensing technology and MEMS micro-sensing technology can better meet the testing needs of the jacked pile penetration mechanism of clay, and accurately monitor the force status of the pile body during the penetration of the model pile. The sensor installation method used in the test is feasible.(2)The pile axial force decreases with the increase in penetration depth, and the decreasing amplitude decreases gradually. There is degradation in the pile side resistance. The difference of pile axial force at the same depth decreases gradually and the degradation degree of pileside resistance decreases with the penetration depth.(3)In the whole penetration process, the ratio of pile end resistance to pile pressure is constantly changing. At the beginning of penetration, the pile end resistance increases rapidly, and its proportion is much higher than the pile lateral resistance. In the middle and late stage of pile penetration, the ratio of pile lateral resistance increases. The growth rate of pile end resistance is relatively stable and its proportion is always high.(4)The radial effective stress in the process of pile jacking in homogeneous soil increases linearly with the penetration depth. The radial effective stress occupies a high proportion in the total radial stress, and the distribution law of the two is consistent. By comparing the data obtained from sensors at different positions, it can be found that the radial effective stress also degenerates at the same depth. In the process of pile jacking in homogeneous soil, the excess pore water pressure fluctuates with the increase in depth. According to the comparison of data at the same depth, it is found that the excess pore water pressure dissipates, and the degradation range decreases with the increase in depth.

## Figures and Tables

**Figure 1 micromachines-11-00876-f001:**
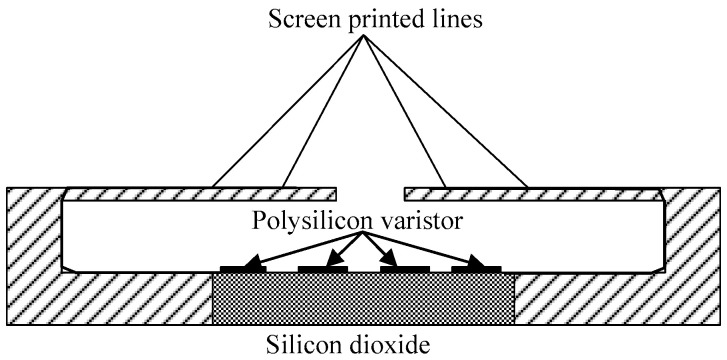
Schematic diagram of silicon piezoresistive pressure sensor.

**Figure 2 micromachines-11-00876-f002:**
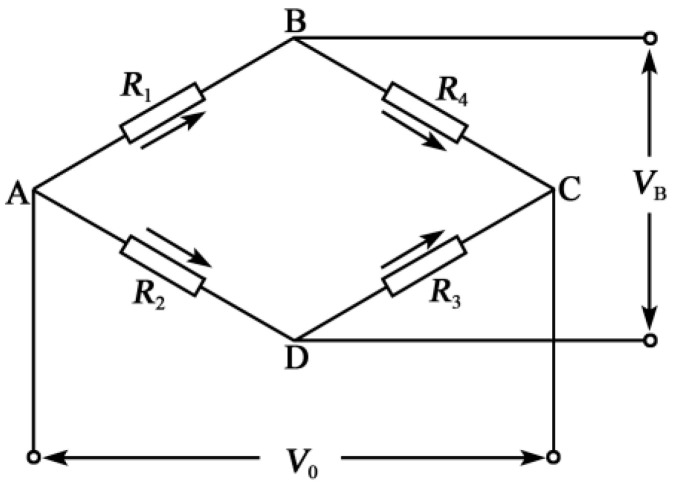
Wheatstone bridge circuit.

**Figure 3 micromachines-11-00876-f003:**
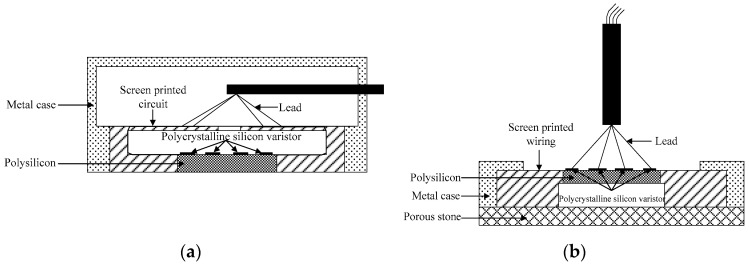
Schematic diagram of miniature silicon piezoresistive pressure sensor package: (**a**) Lateral outgoing earth pressure sensor and (**b**) bottom outgoing pore water pressure sensor.

**Figure 4 micromachines-11-00876-f004:**
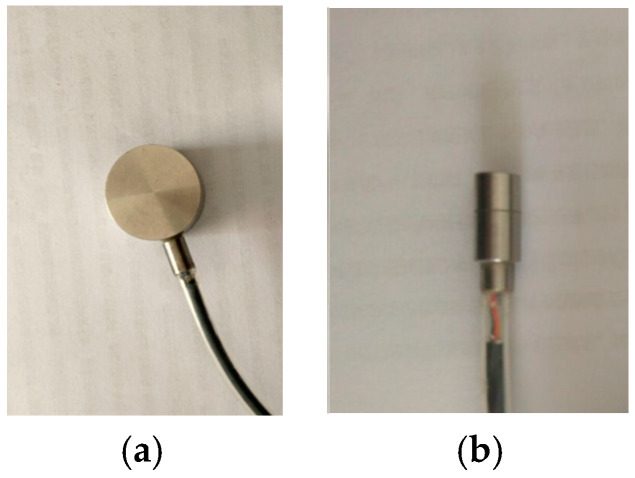
Picture of miniature silicon piezoresistive pressure sensor: (**a**) Earth pressure sensor and (**b**) pore water pressure sensor.

**Figure 5 micromachines-11-00876-f005:**
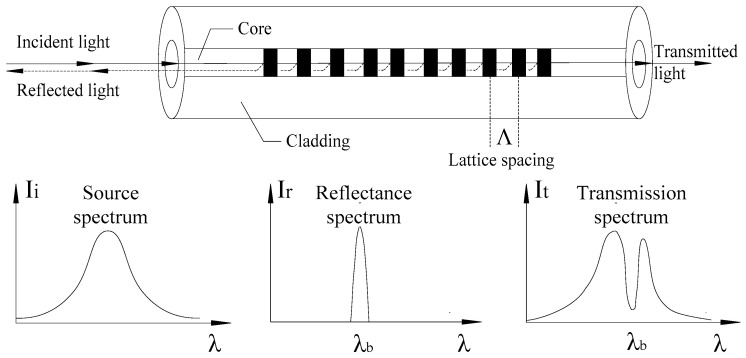
Fiber Bragg grating (FBG) quasi-distributed measurement principle. Where *I_i_* is light source spectrum; *I_r_* is the reflection spectrum; *I_t_* is the conduction spectrum; *λ* is the fiber Bragg grating wavelength (nm); and *λ_b_* is the fiber grating center wavelength (nm).

**Figure 6 micromachines-11-00876-f006:**
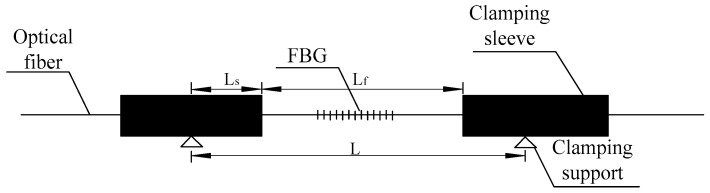
Working principle and structure diagram of sensitized FBG sensor.

**Figure 7 micromachines-11-00876-f007:**
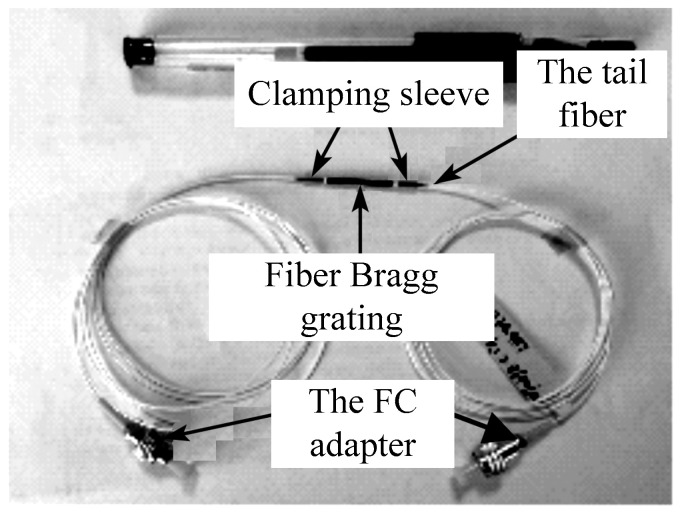
Photo of sensitized miniature FBG strain sensor.

**Figure 8 micromachines-11-00876-f008:**
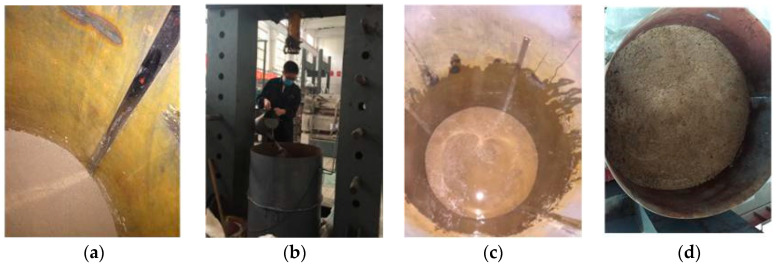
Preparation of foundation soil: (**a**) Layered with soil, (**b**) add water wetting, (**c**) uniform infiltration, and (**d**) rest foundation soil.

**Figure 9 micromachines-11-00876-f009:**
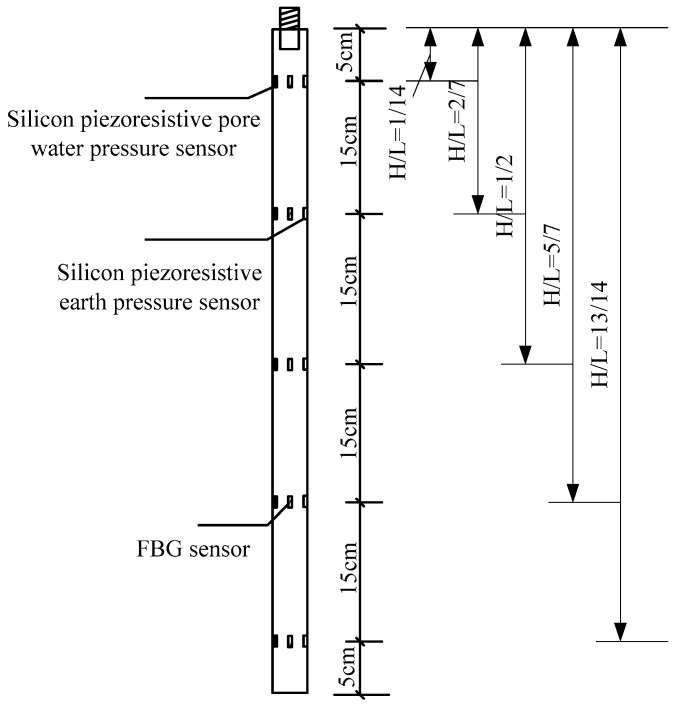
Distribution diagram of sensor arrangement.

**Figure 10 micromachines-11-00876-f010:**
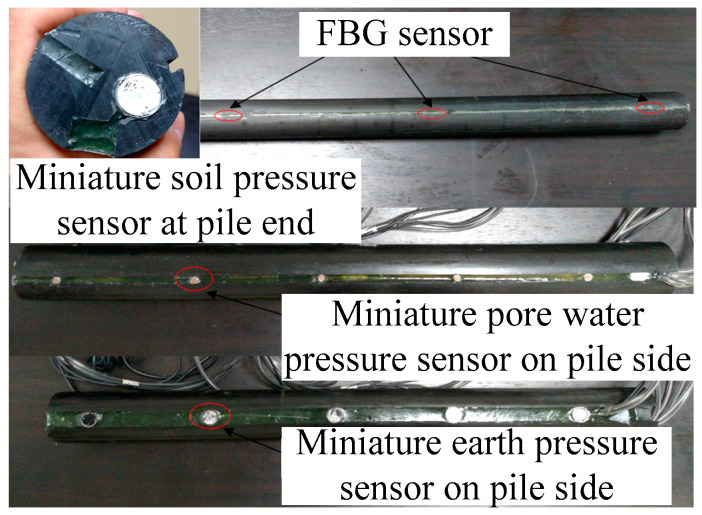
Sensor installation.

**Figure 11 micromachines-11-00876-f011:**
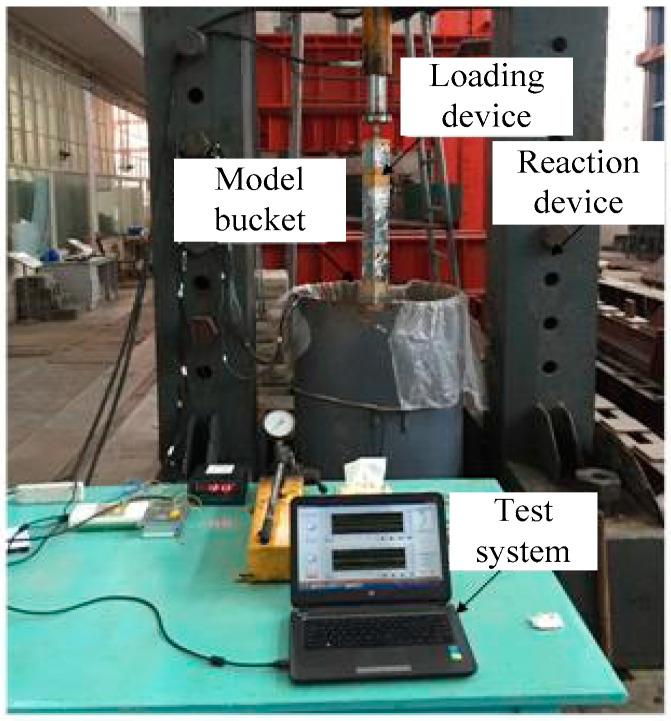
Test device.

**Figure 12 micromachines-11-00876-f012:**
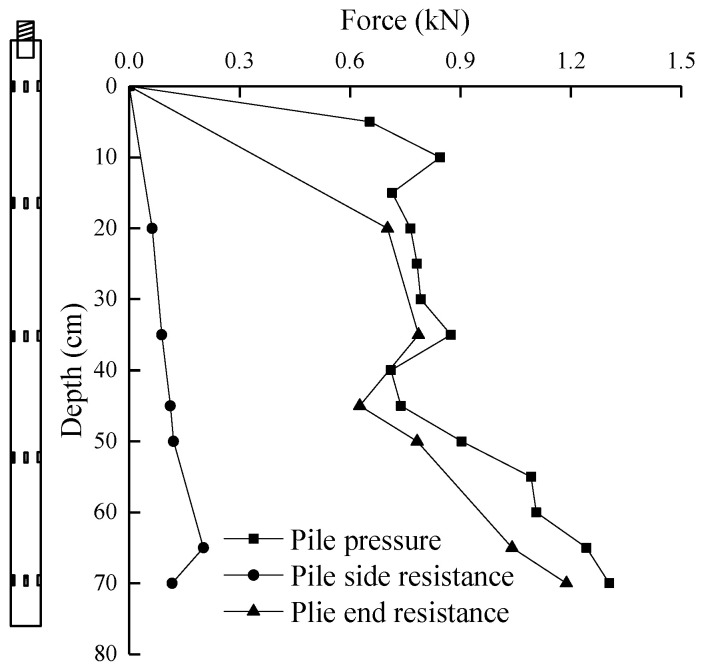
Pile driving force with penetration depth curve.

**Figure 13 micromachines-11-00876-f013:**
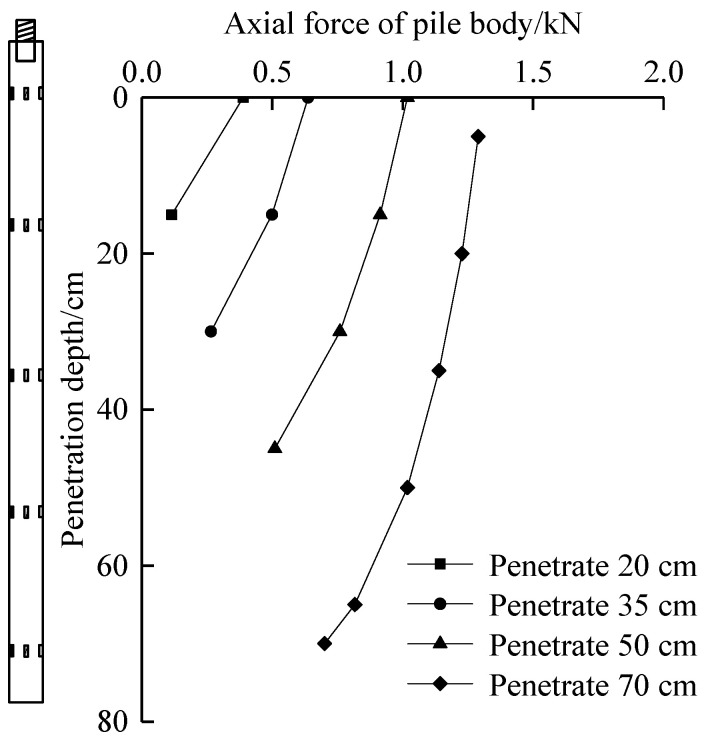
Pile axial force with penetration depth curve.

**Figure 14 micromachines-11-00876-f014:**
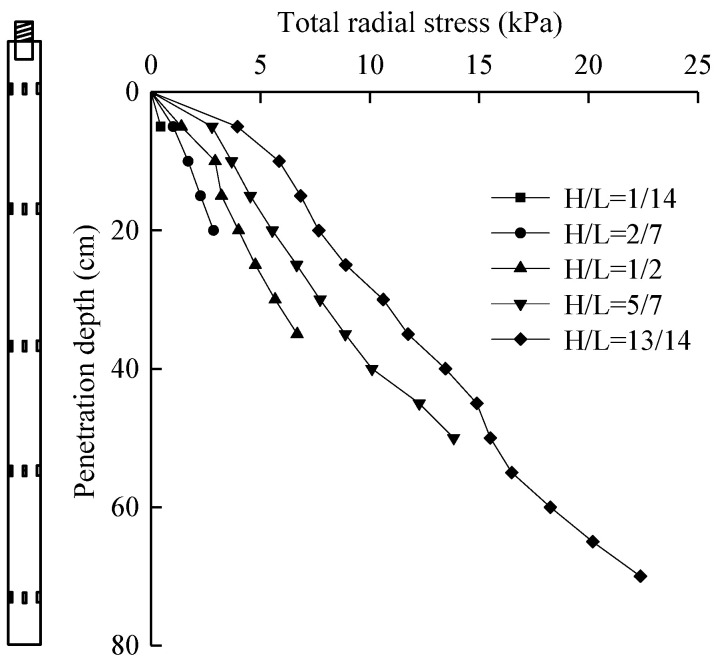
Relationship curve between total radial stress and penetration depth.

**Figure 15 micromachines-11-00876-f015:**
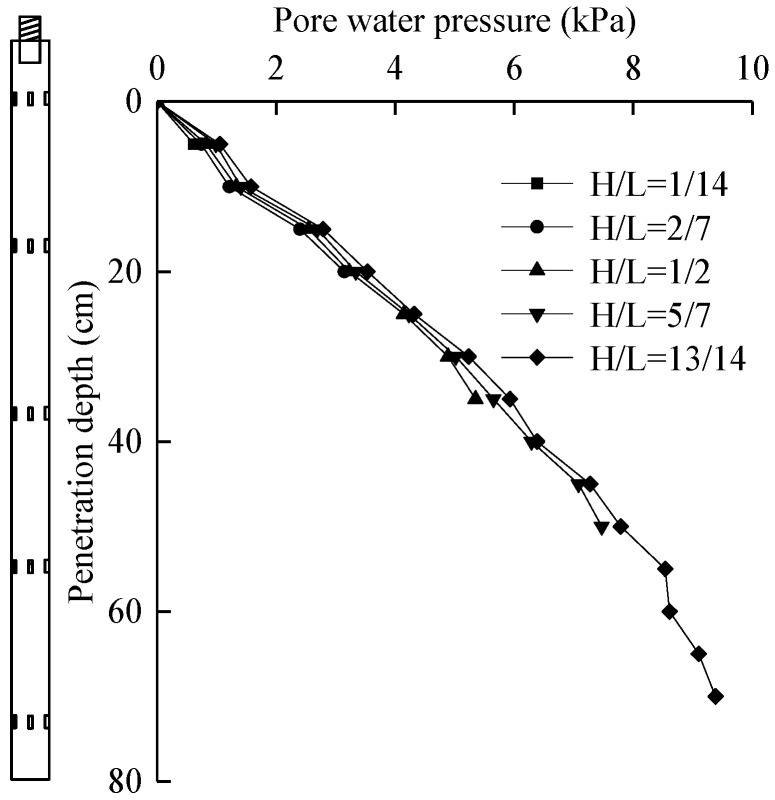
Relationship curve between pore water pressure and penetration depth.

**Figure 16 micromachines-11-00876-f016:**
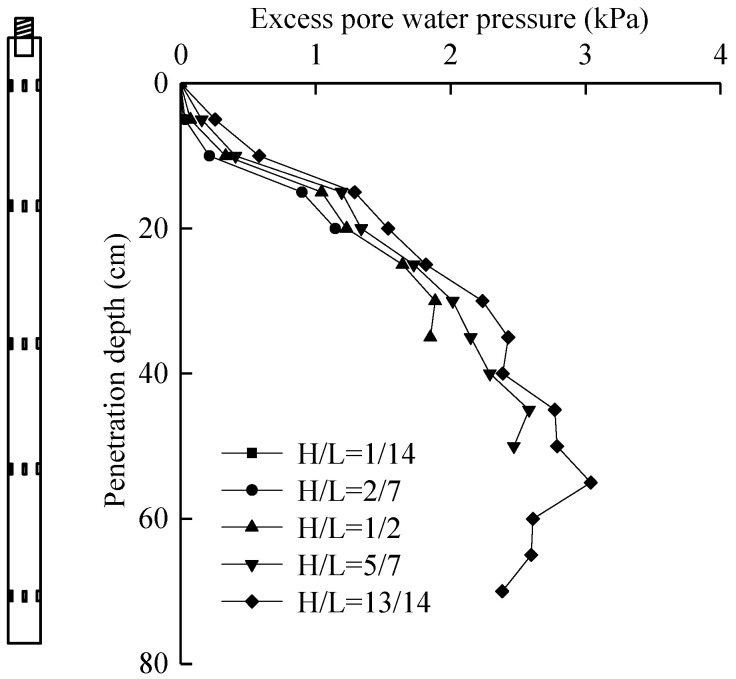
Relationship curve between excess pore water pressure and penetration depth.

**Figure 17 micromachines-11-00876-f017:**
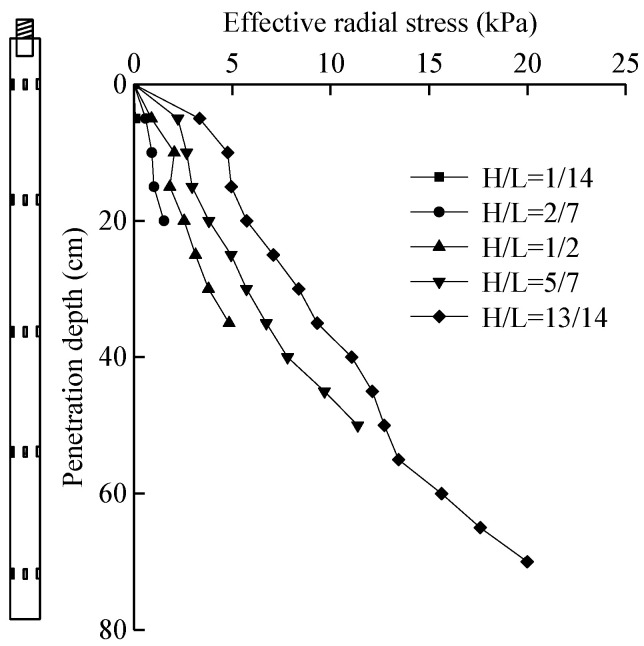
Relationship curve between effective radial stress and penetration depth.

**Table 1 micromachines-11-00876-t001:** Performance indexes of miniature silicon piezoresistive pressure sensor.

Parameter	Earth Pressure Sensor	Pore Water Pressure Sensor
Range (kPa)	500	500
Dynamic response (kHz)	2000	2000
Precision	0.1%	0.1%
Interface means	Built-in locking structure	Built-in locking structure
Appearance of size (mm × mm)	20 × 10	8 × 15

**Table 2 micromachines-11-00876-t002:** Performance indexes of sensitized FBG strain sensor.

Sensor Number	No.1	No.2	No.3
Range (με)	±1000	±1000	±1000
Resolution ratio (με)	1	1	1
Interface means	FC/APC	FC/APC	FC/APC

**Table 3 micromachines-11-00876-t003:** Physical properties of soil.

Density (g/cm^3)^	Moisture Content (%)	Dry density (g/cm^3^)	Relative Density	Liquid Limit (%)	Plastic Limit (%)	Internal Friction Angle (°)	Cohesion (kPa)	Saturability (%)	Void Ratio	Coefficient of Compressibility (MPa^−1^)	Modulus of Compression (MPa)
1.98	25.3	1.58	2.73	31.3	16.5	8.6	14.4	94.9	0.728	0.32	5.5

**Table 4 micromachines-11-00876-t004:** Earth pressure sensor and pore water pressure sensor parameters.

Technical Parameters	Dynamic Response (kHz)	Linearity (%FSO)	Hysteresis (%FSO)	Repeatability (%FSO)	Precision (%)	Interface Means
Numerical value	2000	0.19	0.03	1.03	0.1	Built-in locking structure

**Table 5 micromachines-11-00876-t005:** Ratio of excess pore water pressure to effective overlying soil weight at different depths.

Depth of Sensor (cm)	Excess Pore Water Pressure (kPa)	Effective Overburden Weight (kPa)	Ratio (%)
10	0.582	0.88	66.1
30	2.236	2.64	84.7
50	2.789	4.4	63.4
70	2.382	6.16	38.7

**Table 6 micromachines-11-00876-t006:** Ratio of effective radial stress to total radial stress of model pile at different depths.

Depth of Sensor (cm)	Effective Radial Stress (kPa)	Total Effective Stress (kPa)	Ratio (%)
10	4.773	5.855	81.5
30	8.379	10.615	78.9
50	12.726	15.515	82.0
70	19.993	22.375	89.4

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
