# Peer review of "Application of Miniature FBG-MEMS Pressure Sensor in Penetration Process of Jacked Pile"

_micromachines, 2020, doi:10.3390/mi11090876_

Round 1

Reviewer 1 Report

In this work, the authors report about the application of silicon piezoresistive and fiber Bragg grating (FBG) sensors for the measurement of pressure during penetration process of jacked piles.

The following points have to be addressed in a revised version:

  1. Figures 12, 13, 14, 15, 16, 17 should be enriched with a schematic (it could be similar to Fig. 9, for example) highlighting the sensors involved in each related measurement.
  2. Which is the repeatability of the results and trends found by the authors? Have the authors repeated the experiments and verified the same trends?
  3. The sentences on lines 96-100 are not clear and have to be rephrased.
  4. Are the silicon piezoresistive sensors fabricated by the authors? Differently, please provide model, manufacturer, and details about the interrogation system.
  5. Same as above for FBGs, provide length, manufacturer, and interrogation system.
  6. In Section 4.4 the number of each kind of sensors should be clearly reported.

Additional comments:

  • In abstract first line and on line 72, “jacked” is mistyped.
  • In Eq. 4 it is λB.
  • On line 162, it is mentioned Eq. (11) but there is no such equation in the paper.

Reviewer 2 Report

Some grammatical mistakes must be corrected (line 10 in the abstract jacket instead of jakcet, line 72 kacked???, and others).

Line 140. FBG needs always to be corrected for temperature, even for small changes. How to do it must be implemented. 

Explain why fig 13 is so different in trend from fig 12

Paragraph 5.2 It is said that axial force on the bar is obtained from FBGs measurements ( which are strains). How to correlate from one to the other need to be explained. Also the bar seems to have 4 FBGs, which maybe do not give the same readings, particularly if there are bending effects. Details of the experiment are needed.

A simple model to understand the experimental results would be very useful.

Conclusion 2 repeat the same paragraph .

In summary, paper needs to be written with more detail for the readers

Round 2

Reviewer 1 Report

The revised version of the work is suitable for publication.

Author Response

Thank the reviewer for his comments.

Reviewer 2 Report

My former comments were taken into account,

I have only a new comment, which maybe the authors may consider before paper publication ( without a need for a new submission to reviewers). 

The bar is compressed, so the FBGs have to register negative strains. The authors have proposed an artifact to increase the sensitivity of the FBG (fig 6), but if the optical fiber is unsupported, just bonded at the ends, it can not detect negative strains, because of microbuckling. Please clarify

Author Response

Thank you very much for your comments. In this paper, the optical fiber is supported. The two ends of the fiber Bragg grating were fixed in the gripper tube with binder to avoid the influence of binder on the strain transfer of the FBG. Then the sensor is fixed on the test member through the clamping support. Therefore, the axial negative strain can be obtained through the FBG sensor. At the same time, in order to ensure the test performance of the FBG sensor, we also took a lot of protection measures during the test reading to try to get more accurate data.